# Multi-CryoGAN: Reconstruction of Continuous Conformations in Cryo-EM Using Generative Adversarial Networks

**Abstract.** We propose a deep-learning-based reconstruction method for cryo-electron microscopy (Cryo-EM) that can model multiple conformations of a nonrigid biomolecule in a standalone manner. Cryo-EM produces many noisy projections from separate instances of the same but randomly oriented biomolecule. Current methods rely on pose and conformation estimation which are inefficient for the reconstruction of continuous conformations that carries valuable information. We introduce Multi-CryoGAN, which sidesteps the additional processing by casting the volume reconstruction into the distribution matching problem. By introducing a manifold mapping module, Multi-CryoGAN can learn continuous structural heterogeneity *without pose estimation* nor *clustering*. We also give a theoretical guarantee of recovery of the true conformations. Our method can successfully reconstruct 3D protein complexes on synthetic 2D Cryo-EM datasets for both continuous and discrete structural variability scenarios. Multi-CryoGAN is the first model that can reconstruct continuous conformations of a biomolecule from Cryo-EM images in a fully unsupervised and end-to-end manner.

**Keywords:** Cryo-EM, Inverse problem, Image reconstruction, Generative Adversarial Networks, Continuous protein conformations

## 1 Introduction

The determination of the structure of nonrigid macromolecules is an important aspect of structural biology and is fundamental in our understanding of biological mechanisms and in drug discovery [3]. Among other popular techniques such as X-ray crystallography and nuclear magnetic resonance spectroscopy, Cryo-EM has emerged as a unique method to determine molecular structures at unprecedented high resolutions. It is widely applicable to proteins that are difficult to crystallize or have large structures. Cryo-EM produces a large number (from $10^4$ to $10^7$) of tomographic projections of the molecules dispersed in a frozen solution. The reconstruction of 3D molecular structures from these data involves three main challenges: possible structural heterogeneity of the molecule, random locations and orientations of the molecules in the ice, and an extremely poor signal-to-noise ratio (SNR), which can be as low as to -20 dB (Figure 1). In fact, the reconstruction of continuously varying conformations of a nonrigid molecule

Fig. 1: Reconstruction task of Cryo-EM. Many samples of a biomolecule (which may exhibit continuously varying conformations) are frozen in vitreous ice. These are then imaged/projected using an electron beam to get 2D micrographs. The 2D images containing projection of a single sample are then picked out (black-box). The task then is to reconstruct the conformations of the biomolecule from these measurements.

is still an open problem in the field [6, 17]. A solution would considerably enhance our understanding of the functions and behaviors of many biomolecules.

Most current methods [18, 19] find the 3D structure by maximizing the like-lihood of the data. They employ an iterative optimization scheme that alterna-tively estimates the distribution of poses (or orientations) and reconstructs a 3D structure until a criterion is satisfied (Figure 2(a)). To address the structural variability of protein complexes, these methods typically use discrete clustering approaches. However, the pose estimation and clustering steps are computation-ally heavy and include heuristics. This makes these methods inefficient when the molecule has continuous variations or a large set of discrete conformations.

Recently, two deep-learning-based reconstruction methods that require no prior-training nor additional training data have been introduced. On one hand, CryoDRGN [26] uses a variational auto-encoder (VAE) to model continuous structural variability, avoiding the heuristic clustering step. It is a likelihood-based method that requires pose estimation using an external routine like a branch-and bound-method [18]. This additional processing step can complicate the reconstruction procedure and limit the flexibility of the model. On the other hand, Gupta *et al.* [10] have recently proposed CryoGAN. It addresses the prob-lem under a generative adversarial framework [8]. CryoGAN learns to reconstruct a 3D structure whose randomly projected 2D Cryo-EM images match the ac-quired data in a distributional sense (Figure 2(b)). Due to this likelihood-free characteristic, CryoGAN does not require any additional processing step such as pose estimation, while it can be directly deployed on the Cryo-EM measure-ments. This largely helps simplify the reconstruction procedure. However, its application is limited to the reconstruction of a single conformation.

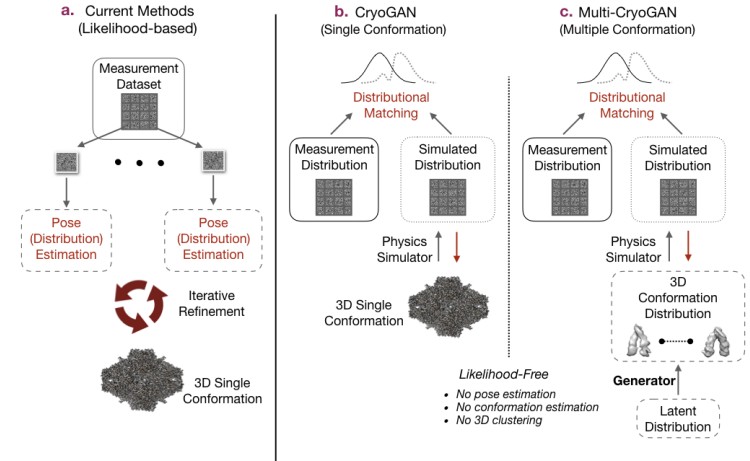

Fig. 2: Schematic overview of the reconstruction methods in Cryo-EM. (a) Current methodels; (b) CryoGAN; (c) proposed method (Multi-CryoGAN).

In this paper, we combine the advantages of CryoDRGN and CryoGAN. We propose an unsupervised deep-learning-based method, called Multi-CryoGAN. It can reconstruct continuously varying conformations of a molecule in a truly standalone and likelihood-free manner. Using a convolutional neural network (CNN), it directly learns a mapping from a latent space to the 3D conformation distribution. Unlike current methods, it requires no calculation of pose or conformation estimation for each projection, while it has the capacity to reconstruct low-dimensional but complicated conformation manifolds [11].

Using synthetic Cryo-EM data as our benchmark, we show that our method can reconstruct the conformation manifold for both continuous and discrete conformation distributions. In the discrete case, it also reconstructs the corresponding probabilities. To the best of our knowledge, this is the first standalone method that can construct whole manifold of the biomolecule conformations.

## 2    Related Work

**Traditional Cryo-EM Image Reconstruction.** A detailed survey of the classical methods is provided in [21, 22]. Most of them fall into the maximum-likelihood (ML) framework and rely on either expectation-maximization (ML-EM) [19] or gradient descent (the first stage of [18]). In the context of heterogeneous conformation reconstruction, a conjugate-gradient descent is used to estimate the volume covariance matrix [1]. The eigendecomposition of this matrix contains information about the conformation distribution which is then input to the ML framework. In [5], a conformation manifold is generated for each group of projections with similar poses. This data-clustering approach assumes

orientation rather than structural heterogeneity to be the dominant cause for variations among the projection images, a strong constraint. In addition, the reconstruction of 3D movies from multiple 2D manifolds can be computationally expensive. In another method, Moscovich *et al.* [16] compute the graph Laplacian eigenvectors of the conformations using covariance estimation. In [14], the problem of heterogeneous reconstructions is reformulated as the search for a homogeneous high-dimensional structure that represents all the states, called a hypermolecule, which is characterized by a basis of hypercomponents. This allows for reconstruction of high-dimensional conformation manifolds but requires assumptions on the variations of the conformations as a prior in their Bayesian formulation.

One of the main drawbacks of these methods is that they require marginalization over the space of poses for each projection image, which is computationally demanding and potentially inaccurate. In addition, because they rely on 3D clustering to deal with structural variations of protein complexes, these methods become inefficient for a large set of discrete conformations and struggle to recover a continuum of conformations.

**Deep Learning for Cryo-EM Reconstructions.** In addition to CryoDRGN and CryoGAN that have already been discussed in the introduction, there is a third described in [15]. It uses a VAE and a framework based on a generative adversarial network (GAN) to learn the latent distribution of the acquired data. This representation is then used to estimate the orientation and other important parameters for each projection image.

**Deep Learning to Recover a 3D Object from 2D Projections.** The implicit or explicit recovery of 3D shapes from 2D views is an important problem in computer vision. Many deep-learning algorithms have been proposed for this [7, 24, 23]. Taking inspiration from compressed sensing, Bora *et al.* [4] have recently introduced a GAN framework that can recover an original distribution from the measurements through a forward model. While these approaches would in principle be applicable, they consider a forward model that is too simple for Cryo-EM, where a contrast transfer function (CTF) must be taken into account and where the noise is orders of magnitude stronger (e.g. with a typical SNR of -10 ot -20 dB.

## 3    Background and Preliminaries

### 3.1    Image-Formation Model

The aim of Cryo-EM is to reconstruct the 3D molecular structure from the measurements $\{\mathbf{y}_{\mathrm{data}}^1, \ldots, \mathbf{y}_{\mathrm{data}}^Q\}$, where $Q$ is typically between $10^4$ to $10^7$. Each measurement $\mathbf{y}^q \in \mathbb{R}^{N \times N}$ is given by

$$\mathbf{y}^q = \underbrace{\mathbf{C}_{\mathbf{c}^q} * \mathbf{S}_{\mathbf{t}^q} \mathbf{P}_{\boldsymbol{\theta}^q}}_{\mathbf{H}_{\boldsymbol{\varphi}^q}} \{\mathbf{x}^q\} + \mathbf{n}^q, \tag{1}$$

where

- $\mathbf{x}^q \in \mathbb{R}^{N \times N \times N}$ is a separate instance of the 3D molecular structure;
- $\mathbf{n}^q \in \mathbb{R}^{N \times N}$ is the noise;
- $\mathbf{H}_{\boldsymbol{\varphi}^q}$ is the measurement operator which depends on the imaging parameters $\boldsymbol{\varphi}^q = (\boldsymbol{\theta}^q, \mathbf{t}^q, \mathbf{c}^q) \in \mathbb{R}^8$ and involves three operations.
    - The term $\mathbf{P}_{\boldsymbol{\theta}^q}\{\mathbf{x}^q\}$ is the tomographic projection of $\mathbf{x}^q$ rotated by $\boldsymbol{\theta}^q = (\theta_1^q, \theta_2^q, \theta_3^q)$.
    - The operator $\mathbf{S}_{\mathbf{t}^q}$ shifts the projected image by $\mathbf{t}^q = (t_1^q, t_2^q)$. This shift arises from off-centered particle picking.
    - The Fourier transform of the resulting image is then modulated by the CTF $\hat{\mathbf{C}}_{\mathbf{c}^q}$ with defocus parameters $\mathbf{c}^q = (d_1^q, d_2^q, \alpha_{\text{ast}}^q)$ and thereafter subjected to inverse Fourier Transform.

For more details, please see *Supplementary: Image Formation* in [10]. The challenge of Cryo-EM is that, for each measurement $\mathbf{y}^q$, the structure $\mathbf{x}^q$ and the imaging parameters $(\boldsymbol{\theta}^q, \mathbf{t}^q)$ are unknown, the CTF is a band pass filter with multiple radial zero frequencies that incur irretrievable loss of information, and the energy of the noise is multiple times ($\sim$10 to 100 times) that of the signal which corresponds to SNRs of -10 to -20 dB. In the homogeneous case (single conformation), all $\mathbf{x}^q$ are identical. But in the heterogeneous case (multiple conformations), each $\mathbf{x}^q$ represents a different conformation of the same biomolecule.

**Stochastic Modeling**. We denote the probability distribution over the conformation landscape by $p_{\text{conf}}(\mathbf{x})$ from which a conformation $\mathbf{x}^q$ is assumed to be sampled from. We assume that the imaging parameters and the noise are sampled from known distributions $p_{\boldsymbol{\varphi}} = p_{\boldsymbol{\theta}} p_{\mathbf{t}} p_{\mathbf{c}}$ and $p_{\mathbf{n}}$, respectively. For a given conformation distribution $p_{\text{conf}}(\mathbf{x})$, this stochastic forward model induces a distribution over the measurements which we denote by $p(\mathbf{y})$. We denote by $p_{\text{conf}}^{\text{data}}(\mathbf{x})$ the true conformation distribution from which the data distribution $p_{\text{data}}(\mathbf{y})$ is acquired such that $\{\mathbf{y}_{\text{data}}^1, \ldots, \mathbf{y}_{\text{data}}^Q\} \sim p_{\text{data}}(\mathbf{y})$. The distribution $p_{\text{conf}}^{\text{data}}(\mathbf{x})$ is unknown and needs to be recovered.

The classical methods are likelihood-based and rely on the estimation of imaging parameters $(\boldsymbol{\theta}^q, \mathbf{t}^q)$ (or a distribution over them) and of the conformation class for each measurement image $\mathbf{y}^q$. This information is then utilized to reconstruct the multiple discrete conformations. Our method, in contrast, is built upon the insight that, to recover $p_{\text{conf}}^{\text{data}}(\mathbf{x})$ it is sufficient to find a $p_{\text{conf}}^{\text{gen}}(\mathbf{x})$ whose corresponding measurement distribution $p_{\text{gen}}(\mathbf{y})$ is equal to $p_{\text{data}}(\mathbf{y})$ (see Theorem 1). This does away with pose (or distributions over the poses) estimation and conformation clustering for each measurement.

### 3.2   CryoGAN

Our scheme is extension of the CryoGAN [10] method, which is applicable only for the homogeneous case $p_{\text{conf}}^{\text{data}}(\mathbf{x}) = \delta(\mathbf{x} - \mathbf{x}_{\text{data}})$, where $\mathbf{x}_{\text{data}}$ is the true 3D structure. CryoGAN tackles the challenge by casting the reconstruction problem as a distribution-matching problem (Figure 2(b)). More specifically, it learns to reconstruct the 3D volume $\mathbf{x}^*$ whose simulated projection set (measurement

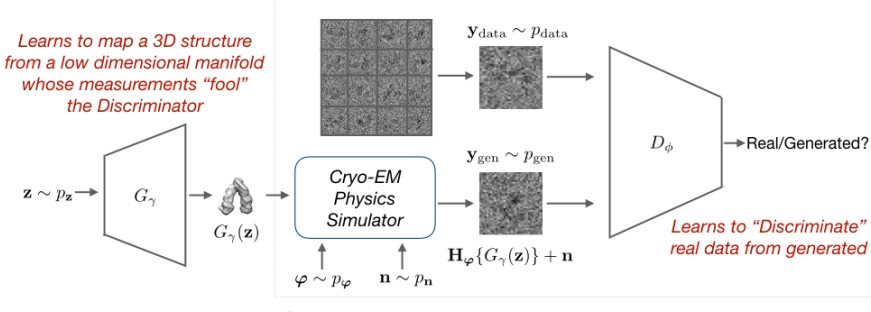

Fig. 3: Schematic illustration of Multi-CryoGAN and its components. (a) Conformation manifold mapper; (b) CryoGAN.

distribution) is most similar to the real projection data in a distributional sense, such that

$$\mathbf{x}^* = \arg\min_{\mathbf{x}} \mathrm{WD}(p_{\mathrm{data}}(\mathbf{y})||p_{\mathrm{gen}}(\mathbf{y};\mathbf{x})). \quad (2)$$

Here, $p_{\mathrm{gen}}(\mathbf{y};\mathbf{x})$ is the distribution generated from the Cryo-EM physics simulator and WD refers to the Wasserstein distance [25]. This goal is achieved by solving the min-max optimization problem:

$$\mathbf{x}^* = \arg\min_{\mathbf{x}} \underbrace{\max_{D_\phi:\|D_\phi\|_L \leq 1} (\mathbb{E}_{\mathbf{y}\sim p_{\mathrm{data}}(\mathbf{y})}[D_\phi(\mathbf{y})] - \mathbb{E}_{\mathbf{y}\sim p_{\mathrm{gen}}(\mathbf{y};\mathbf{x})}[D_\phi(\mathbf{y})])}_{\mathrm{WD}(p_{\mathrm{data}}(\mathbf{y})||p_{\mathrm{gen}}(\mathbf{y};\mathbf{x}))}, \quad (3)$$

where $D_\phi$ is a neural network with parameters $\phi$ that is constrained to have Lipschitz constant $\|D_\phi\|_L \leq 1$ [9] (Figure 3(b)). Here, $D_\phi$ learns to differentiate between the real projection $\mathbf{y}$ and the simulated projection $\mathbf{H}_\varphi\{\mathbf{x}\}$ and scores the realness of given samples. As the discriminative power of $D_\phi$ becomes stronger (`maximization`) and the underlying volume estimate $\mathbf{x}$ is updated accordingly (`minimization`), $p_{\mathrm{data}}(\mathbf{y})$ and $p_{\mathrm{gen}}(\mathbf{y};\mathbf{x})$ become indistinguishable so that the algorithm recovers $\mathbf{x}^* = \mathbf{x}_{\mathrm{data}}$.

## 4    Method

### 4.1    Parameterization of the Conformation Manifold

CryoGAN successfully reconstructs the volumetric structure of a protein by finding a single volume $\mathbf{x}$ that explains the entire set of projections, which is adequate when all the imaged particles are identical (homogeneous case). However, in reality, many biomolecules have nonrigid structures, which carry vital information. To address this, we introduce a manifold-learning module $G_\gamma$ that uses

**Algorithm 1:** Samples from the generated distribution $p_{\text{gen}}(\mathbf{y}; G_\gamma)$.

> **Input:** latent distribution $p_{\mathbf{z}}$; angle distribution $p_{\boldsymbol{\theta}}$; translation distribution $p_{\mathbf{t}}$; CTF parameters distribution $p_{\mathbf{c}}$; noise distribution $p_{\mathbf{n}}$
>
> **Output:** Simulated projection $\mathbf{y}_{\text{gen}}$
>
> 1. Sample $\mathbf{z} \sim p_{\mathbf{z}}$.
> 2. Feed $\mathbf{z}$ into generator network to get $\mathbf{x} = G_\gamma(\mathbf{z})$.
> 3. Sample the imaging parameters $\boldsymbol{\varphi} = [\boldsymbol{\theta}, \mathbf{t}, \mathbf{c}]$.
>    - Sample the Euler angles $\boldsymbol{\theta} = (\theta_1, \theta_2, \theta_3) \sim p_{\boldsymbol{\theta}}$.
>    - Sample the 2D shifts $\mathbf{t} = (t_1, t_2) \sim p_{\mathbf{t}}$.
>    - Sample the CTF parameters $\mathbf{c} = (d_1, d_2, \alpha_{\text{ast}}) \sim p_{\mathbf{c}}$.
>
> 4. Sample the noise $\mathbf{n} \sim p_{\mathbf{n}}$.
> 5. Generate $\mathbf{y}_{\text{gen}} = \mathbf{H}_{\boldsymbol{\varphi}}\mathbf{x} + \mathbf{n}$ based on (8).
>
> **return** $\mathbf{y}_{\text{gen}}$

a CNN with learnable weights $\gamma$ (Figure 3 (a)). Sampling from $p_{\text{conf}}(\mathbf{x})$ is then equivalent to getting $G_\gamma(\mathbf{z})$, where $\mathbf{z}$ is sampled from a prior distribution. Therefore, $\mathbf{y} \sim p_{\text{gen}}(\mathbf{y})$ is obtained by evaluating $\mathbf{H}_{\boldsymbol{\varphi}}\{G_\gamma(\mathbf{z})\} + \mathbf{n}$, where $(\mathbf{n}, \mathbf{z}, \boldsymbol{\varphi})$ are sampled from their distributions (see Algorithm 1 and Figure 3). To explicitly show this dependency to $G_\gamma$, we hereafter denote the generated distribution of projection data by $p_{\text{gen}}(\mathbf{y}; G_\gamma)$.

### 4.2 Optimization Scheme

We now find $G_{\gamma^*}$ such that the distance between $p_{\text{data}}(\mathbf{y})$ and $p_{\text{gen}}(\mathbf{y}; G_\gamma)$ is minimized, which results in the min-max optimization problem [2]

$$G_{\gamma^*} = \arg\min_{G_\gamma} \text{WD}(p_{\text{data}}(\mathbf{y}) || p_{\text{gen}}(\mathbf{y}; G_\gamma)) \tag{4}$$

$$= \arg\min_{G_\gamma} \underbrace{\max_{D_\phi : \|D_\phi\|_L \leq 1} \left( \mathbb{E}_{\mathbf{y} \sim p_{\text{data}}}[D_\phi(\mathbf{y})] - \mathbb{E}_{\mathbf{y} \sim p_{\text{gen}}(\mathbf{y}; G_\gamma)}[D_\phi(\mathbf{y})] \right)}_{\text{WD}(p_{\text{data}}(\mathbf{y}) || p_{\text{gen}}(\mathbf{y}; G_\gamma))}. \tag{5}$$

As will be discussed in Theorem 1, the global minimizer $G_{\gamma^*}$ of (5) indeed captures the true conformation landscape $p_{\text{conf}}(\mathbf{x})$, which is achieved when $D_\phi$ is no longer able to differentiate the samples from $p_{\text{data}}(\mathbf{y})$ and $p_{\text{gen}}(\mathbf{y}; G_{\gamma^*})$.

It is crucial to note the difference between Multi-CryoGAN and conventional generative adversarial frameworks [8]. In the latter, $G$ directly outputs the samples from $p_{\text{gen}}(\mathbf{y})$, whereas ours outputs the samples $\mathbf{x}$ from the conformation distribution $p_{\text{conf}}(\mathbf{x})$ whose stochastic projections are the samples of $p_{\text{gen}}(\mathbf{y})$. The conventional schemes only helps one to generate samples which are similar to the real data but does not recover the underlying conformation landscape. Our proposed scheme includes the physics of Cryo-EM, which ties $p_{\text{gen}}(\mathbf{y})$ with the conformation landscape $p_{\text{conf}}(\mathbf{x})$ and is thus able to recover it (See Theorem 1 in Section 6).

**Algorithm 2:** Reconstruction of multiple conformations using Multi-CryoGAN.

**Input:** Dataset $\{\mathbf{y}_{\text{data}}^1, \ldots, \mathbf{y}_{\text{data}}^Q\}$; training parameters: number of steps $k$ to apply to the discriminator and penalty parameter $\lambda$

**Output:** A mapping $G$ from the latent space to the 3D conformation space.

**for** $n_{\text{train}}$ training iterations **do**

  **for** $k$ steps **do**

    – sample from real data: $\{\mathbf{y}_{\text{data}}^1, \ldots, \mathbf{y}_{\text{data}}^B\}$.

    – sample from generated data: $\{\mathbf{y}_{\text{gen}}^1, \ldots, \mathbf{y}_{\text{gen}}^B\} \sim p_{\text{gen}}(\mathbf{y}; G_\gamma)$ (see Algorithm 1).

    – sample from $\{\kappa_1, \ldots, \kappa_B\} \sim \mathrm{U}[0, 1]$.

    – compute $\mathbf{y}_{\text{int}}^b = \kappa_b \cdot \mathbf{y}_{\text{batch}}^b + (1 - \kappa_b) \cdot \mathbf{y}_{\text{gen}}^b$ for all $b \in \{1, \ldots, B\}$.

    – update the discriminator $D_\phi$ by gradient ascent on the loss (5) complemented with the gradient penalty term from [9].

  **end**

  – sample generated data: $\{\mathbf{y}_{\text{gen}}^1, \ldots, \mathbf{y}_{\text{gen}}^B\} \sim p_{\text{gen}}(\mathbf{y}; G_\gamma)$ (see Algorithm 1).

  – update the Generator $G_\gamma$ by gradient descent on the loss (5).

**end**

**return** $\mathbf{G}_{\gamma^*}$

## 5    Experiments and Results

We evaluate the performance of the proposed algorithm on synthetic datasets obtained from a protein with multiple conformations. We synthesize two datasets: one continuum of configurations and one where the particles can only take a discrete number of states. During reconstruction, no assumption is made of their continuous or discrete nature, which suggests that our method is capable of learning different conformation distribution behaviors.

**Dataset.** For each dataset, we generate 100,000 simulated projections from the *in vivo* conformation variation of the heat-shock protein *Hsp90*. The Coulomb density maps of each conformation are created by the code provided in [20] with slight modifications. The conformation variation of this protein is represented by the bond-angle $\beta$, which describes the work cycle of the molecule, where the two subunits continuously vary between fully closed ($\beta = 0°$, protein database entry *2cg9*) and fully opened ($\beta = 20°$). We sample $\beta \sim \text{Uniform}\,(0°, 20°)$ for the continuous case and $\beta \sim 20° * \text{Bernoulli}\,(0.75)$ for the discrete case. Here, $\text{Uniform}\,(a, b)$ is the uniform distribution between $a$ and $b$, and $\text{Bernoulli}\,(p)$ denotes the Bernoulli distribution with parameter $p$. A conformation is generated with $(32 \times 32 \times 32)$ voxels, where the size of each voxel is 5 Å. A 2D projection with random orientation of this conformation is obtained ($(32 \times 32)$ image, Figure 4b). The orientation is sampled from a uniform distribution over $SO(3)$. Then, the CTF is applied to this projection image with a defocus uniformly sampled between $[1.0\ \mu\text{m}, 2.0\ \mu\text{m}]$, assuming that the horizontal and vertical defocus

values are the same and there is no astigmatism. Translations/shifts are disabled in these experiments. Finally, Gaussian noise was added to the CTF-modulated images, resulting in an SNR of approximately -10 dB.

**Implementation Details.** The reconstruction of the conformation is done by solving (5) using Algorithm 2. For both continuous and discrete conformations, we use the same distribution $p_{\mathbf{z}} \sim \text{Uniform}(\mathbf{z}_0, \mathbf{z}_1)$, where $\mathbf{z}_0, \mathbf{z}_1 \in \mathbb{R}^{32 \times 32 \times 32}$ are randomly chosen from $\text{Uniform}(0, 0.025)$ and fixed throughout the process. Thus, we do not impose any prior knowledge whether the landscape is continuous or discrete. As we shall see later, this latent distribution is sufficiently rich to represent the variation of interest in the synthetic datasets. The architecture of $D$, $G$, and training details are provided in the supplementary material.

**Metric.** We deploy two metrics based on Fourier-Shell Correlation (FSC). The FSC between two structures $\mathbf{x}_1$ and $\mathbf{x}_2$ is given by

$$\text{FSC}(\omega, \mathbf{x}_1, \mathbf{x}_2) = \frac{\langle \mathbf{V}_{\hat{\mathbf{x}}_1}^{\omega}, \mathbf{V}_{\hat{\mathbf{x}}_2}^{\omega} \rangle}{\|\mathbf{V}_{\hat{\mathbf{x}}_1}^{\omega}\| \|\mathbf{V}_{\hat{\mathbf{x}}_2}^{\omega}\|} \tag{6}$$

where $\mathbf{V}_{\hat{\mathbf{x}}}^{\omega}$ is the vectorization of the shell of $\hat{\mathbf{x}}$ at radius $\omega$ and $\hat{\mathbf{x}}$ is the 3D Fourier transform of $\mathbf{x}$. As first metric, we use the FSC between a reconstructed conformation and the corresponding ground truth conformation. This metric encapsulates the structural quality of an individual reconstructed conformation.

To evaluate the landscape of the conformations, we propose a second metric that we call the matrix $\mathbf{M} \in \mathbb{R}^{L \times L}$ of FSC cross conformations (FSCCC). Its entries are given by

$$\mathbf{M}[m, n] = \text{AreaFSC}(\mathbf{x}_m, \mathbf{x}_n) = \int_{0 < \omega \leq \omega_c} \text{FSC}(\omega, \mathbf{x}_m, \mathbf{x}_n) d\omega \tag{7}$$

where $\mathbf{x}_m$ and $\mathbf{x}_n$ are samples in the reconstructed conformation manifold and $\omega_c$ is the normalized Nyquist frequency. We determine it for the reconstructed landscape by setting $\mathbf{x}_m = G_{\gamma^*}((1 - \alpha_m) * \mathbf{z}_0 + \alpha_m * \mathbf{z}_1)$ where $\alpha_m = (m/L)$ for $m \in \{0, \ldots, L\}$. The matrix $\mathbf{M}$ encapsulates how similar $\mathbf{x}_m$ is compared to other structures $\mathbf{x}_n$ across the manifold ($\mathbf{M}[m, n]$ is proportional to the similarity between $\mathbf{x}_m$ and $\mathbf{x}_n$), hence allowing for a visualization of the manifold.

For the continuous conformation, it is useful to compare the FSCCC of our reconstructions with that of the ground truth. To that end, we also evaluate $\mathbf{M}[m, n]$ when $\mathbf{x}_m$ corresponds to the bond-angle $\beta = 20°(m/L)$, where $m \in \{0, \ldots, L\}$. In our experiments, we used $L = 20$ for all FSCCC calculations.

## 5.1   Continuous Conformations

We give in Figure 4 a qualitative comparison between the ground truth conformation variation, as the angle $\beta$ goes from $0°$ to $20°$, and the reconstructions $G(\gamma^*)(\mathbf{z})$, where $\mathbf{z} = (1 - \alpha)\mathbf{z}_0 + \mathbf{z}_1$ and $\alpha$ goes from 0 to 1. Our method successfully reconstructs a manifold that exhibits smooth continuous conformation variation (Figure 4(a)), where the input parameter $\alpha$ has direct control over

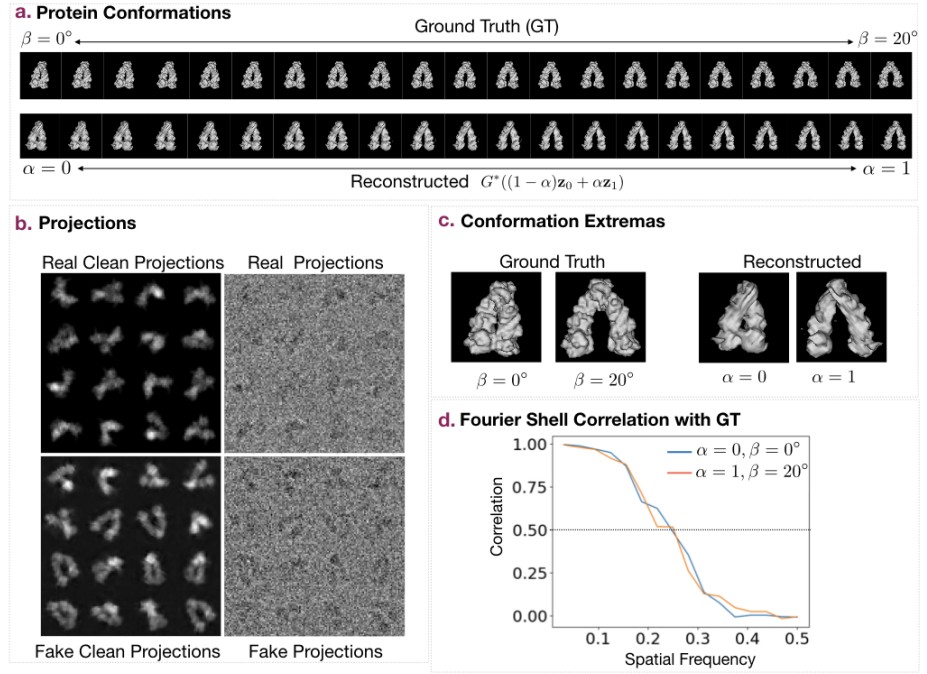

Fig. 4: Continuous conformations experiment. (a) Comparison between the ground truth conformation manifold and the reconstructed conformation manifold $G^*(\mathbf{z})$ where $\mathbf{z} = (1 - \alpha)\mathbf{z}_0 + \alpha\mathbf{z}_1$, $\alpha \in [0, 1]$. (b, left-column) Clean projections of random samples from the ground truth and reconstructed manifold; (b, right-column) their CTF-modulated and noise-corrupted projection. These are the real and generated samples that are fed to $D_\phi$. (c) Ground truth with angles $0°$ and $20°$, and the reconstruction corresponding to the endpoints in the latent space. (d) The FSC between them.

the bond-angle for the reconstruction. This shows that not only the true conformation landscape has been captured, but its factor of variation has been meaningfully encoded by the latent variables $\mathbf{z}$. The similarity between simulated projections and the ground truth data in Figure 4(b) suggests that the algorithm has achieved $p_{\text{data}} = p_{\text{gen}}$. Moreover, their underlying distributions of noiseless projections are also similar, in accordance to the property discussed in Theorem 1.

We also evaluate the structural quality of reconstruction for certain representative individual conformations. In Figure 4(c), the extreme conformations for the ground truth $\beta = 0°$ and $\beta = 20°$ and the reconstructions $\alpha = 0$ and $\alpha = 1$ are shown. Their FSC plot reach the value 0.5 after the normalized frequency of 0.25, so that at least half of the Nyquist resolution is achieved (Figure 4(d)). All these results are further confirmed by the very similar FSCCC ma-

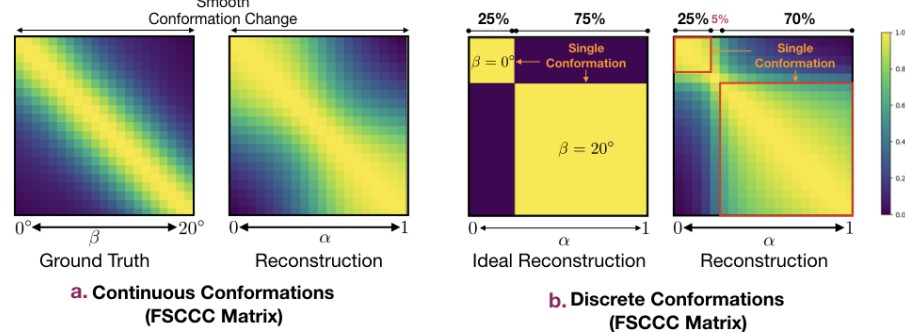

Ground Truth          Reconstruction

**a. Continuous Conformations
(FSCCC Matrix)**

Ideal Reconstruction          Reconstruction

**b. Discrete Conformations
(FSCCC Matrix)**

Fig. 5: The FSC cross-conformation (FSCCC) matrix in (7). (a) Continuous conformation with (left) ground truth and (right) reconstruction. It shows that the reconstructed conformations smoothly vary (without forming clusters) similar to the ground truth case. (b) Discrete conformation case with (left) ideal reconstruction and (right) obtained reconstruction. The ideal reconstruction describes the case where 25% and 75% of latent space would have mapped to the two distinct conformations without any transitions. The obtained reconstruction case can be seen to be very similar to the ideal case with 25% and 70% being the latent space occupied by the two conformations.

trix for ground truth and reconstruction in Figure 5(a). This implies that the reconstruction manifold successfully approximates the continuous ground truth.

## 5.2   Discrete Conformations

We present in Figure 6 the reconstruction results for the discrete case, where our proposed method successfully recovers not only the conformations but also their probabilities. About 70% of the reconstructed landscape matches the configuration for $\beta = 20°$, while 25% matches $\beta = 0°$. The remaining 5% of the landscape corresponds to a relatively abrupt transition between them. This suggests that our model distribution $p_{\text{conf}}(\mathbf{x})$ closely follows the ground truth Bernoulli distribution. This is further supported in Figure Figure 5(b), where the reconstructed FSCCC matrix greatly resembles the ideal reconstruction case. Ideally, one would expect the two conformations to occupy 25% and 75% of the latent space without having any transition conformations. The structural quality of these two recovered configurations with respect to the corresponding ground truth are given in Figure 6(b). Their FSC show that at least half of the Nyquist resolution is achieved (Figure 6(c)).

The FSCCC reconstruction matrix (Figure 5(b)) validates the fact that the reconstructed structures cluster into two main conformations. We use it to determine the probabilities of these cluster/conformations. We determine the probability of a conformation using its first and last row (similarity of the conformations with respect to the extreme conformations). We consider that a con-

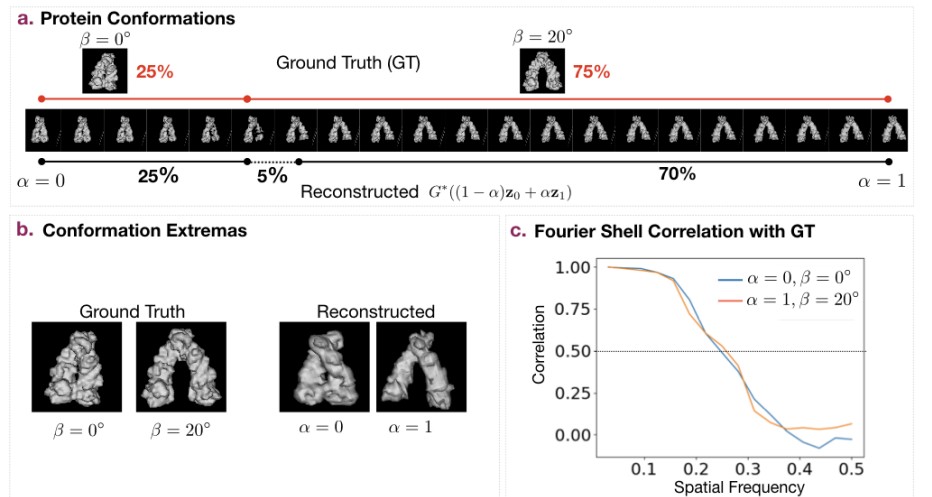

Fig. 6: Discrete conformations experiment. (a) Comparison between the ground truth (GT) taking only two conformations with probability 0.25 and 0.75 and the reconstructed conformations $G_{\gamma^*}(\mathbf{z})$, where $\mathbf{z} = (1 - \alpha)\mathbf{z}_0 + \alpha\mathbf{z}_1$, $\alpha \in [0,1]$. (b) The GT with bond angles 0° and 20° and their reconstruction. (c) FSC of the structures in (b).

formation $\mathbf{x}_n$ belongs to the first cluster/conformation if $\mathbf{M}[0, n] > 0.5$ and $\mathbf{M}[20, n] < 0.5$. If the case is reversed ($\mathbf{M}[0, n] < 0.5$ and $\mathbf{M}[20, n] > 0.5$), then it belongs to the second cluster/conformation. Otherwise, it is considered as a transitioning conformation. This yields that the first 25% and the last 70% structures cluster together to form the $\beta = 0°$ and $\beta = 20°$ conformations, respectively, and the middle 5% are the transitioning conformations.

## 6    Theoretical Guarantee of Recovery

Our experiments illustrate that enforcing a match between the distribution of the simulated measurements and that of the data is sufficient to reconstruct the true conformations. We now prove this mathematically. For the homogeneous case, the proof is already discussed in [10, Theorem 1] which we now extend to the heterogeneous case. We switch to a continuous-domain formulation of the Cryo-EM problem while noting that the result is transferable to the discrete-domain as well, albeit with some discretization error.

**Notations and Preliminaries.** We denote by $\mathcal{L}_2(\mathbb{R}^3)$ the space f 3D structures $f : \mathbb{R}^3 \to \mathbb{R}$ with finite energy $\|f\|_{L_2} < \infty$. The imaging parameters $\boldsymbol{\varphi}$ are assumed to lie in $\mathcal{B} \subset \mathbb{R}^8$. We denote by $\mathcal{L}_2(\mathbb{R}^2)$ the space of 2D measurements with finite energy. Each individual continuous-domain Cryo-EM measurement

$y \in \mathcal{L}_2(\mathbb{R}^2)$ is given by

$$y = H_{\boldsymbol{\varphi}}\{f\} + n, \tag{8}$$

where $f \in \mathcal{L}_2(\mathbb{R}^3)$ is some conformation of the biomolecule sampled from the probability measure $\tilde{\mathbb{P}}_{\mathrm{conf}}$ on $\mathcal{L}_2(\mathbb{R}^3)$, the imaging parameters $\boldsymbol{\varphi}$ are sampled from $p_{\boldsymbol{\varphi}}$, and $n$ is sampled from the noise probability measure $\mathbb{P}_n$ on $\mathcal{L}_2(\mathbb{R}^2)$.

We define $[f] := \{r_{\mathbf{A}}\{f\} : r_{\mathbf{A}} \in O\}$ as the set of all the rotated-reflected versiond of $f$. There, $O$ is the set of all the rotated-reflected versions over $\mathcal{L}_2(\mathbb{R}^3)$. We define the space $\sum \mathcal{L}_2(\mathbb{R}^3) = \mathcal{L}_2(\mathbb{R}^3)/O$ as the quotient space of the shapes. For any $\tilde{\mathbb{P}}_{\mathrm{conf}}$ defined over $\mathcal{L}_2(\mathbb{R}^3)$, an equivalent $\mathbb{P}_{\mathrm{conf}}$ exists over $\sum \mathcal{L}_2(\mathbb{R}^3)$. Since we are interested only in the shape of conformations of the biomolecule, we will only focus on recovering $\mathbb{P}_{\mathrm{conf}}$. We denote by $\Psi$ the probability measure on $\mathcal{B} \in \mathbb{R}^8$. The measure $\Psi$ is associated to the density function $p_{\boldsymbol{\varphi}}$. Both of these induce a probability measure $\mathbb{P}_{\mathrm{clean}}$ on the space $\mathcal{L}_2^2 = \{f : \mathbb{R}^3 \to \mathbb{R}^2 \text{ s.t. } \|f\|_{L_2} < \infty\}$ through the forward operator. This is given by $\mathbb{P}_{\mathrm{clean}}[A] = (\mathbb{P}_{\mathrm{conf}} \times \Psi)[([f], \boldsymbol{\varphi}) \in (\sum \mathcal{L}_2 \times \mathcal{B}) : H_{\boldsymbol{\varphi}} f \in A]$ for any Borel measure set $A \in \mathcal{L}_2(\mathbb{R}^2)$. We denote $\mathbb{P}_{\mathrm{meas}}$ as the probability measure of the noisy measurements.

**Theorem 1.** *Let* $\mathbb{P}_{\mathrm{conf}}^{\mathrm{data}}$ *and* $\mathbb{P}_{\mathrm{conf}}^{\mathrm{gen}}$ *be the true and the reconstructed conformation probability measures on the quotient space of 3D structures* $\sum \mathcal{L}_2(\mathbb{R}^3)$, *respectively. We assume that they are atomic and that they are supported only on nonnegative-valued shapes. Let* $\mathbb{P}_{\mathrm{meas}}^{\mathrm{data}}$ *and* $\mathbb{P}_{\mathrm{meas}}^{\mathrm{gen}}$ *be the probability measures of the noisy Cryo-EM measurements obtained from* $\mathbb{P}_{\mathrm{conf}}^{\mathrm{data}}$ *and* $\mathbb{P}_{\mathrm{conf}}^{\mathrm{gen}}$, *respectively.*

*Make the following physical assumptions:*

1. *the noise probability measure* $\mathbb{P}_n$ *is such that its characteristic functional vanishes nowhere in its domain and that its sample* $n$ *is pointwise-defined everywhere;*
2. *the distributions* $p_{\boldsymbol{\theta}}$, $p_{\mathbf{t}}$, *and* $p_{\mathbf{c}}$ *are bounded;*
3. *for any two* $\mathbf{c}_1, \mathbf{c}_2 \sim p_{\mathbf{c}}, \mathbf{c}_1 \neq \mathbf{c}_2$, *the CTFs* $\hat{\mathbf{C}}_{\mathbf{c}_1}$ *and* $\hat{\mathbf{C}}_{\mathbf{c}_2}$ *share no common zero frequencies.*

*Then, it holds that*

$$\mathbb{P}_{\mathrm{meas}}^{\mathrm{data}} = \mathbb{P}_{\mathrm{meas}}^{\mathrm{gen}} \Rightarrow \mathbb{P}_{\mathrm{conf}}^{\mathrm{data}} = \mathbb{P}_{\mathrm{conf}}^{\mathrm{gen}}. \tag{9}$$

*Proof.* We first prove that $\mathbb{P}_{\mathrm{meas}}^{\mathrm{data}} = \mathbb{P}_{\mathrm{meas}}^{\mathrm{gen}} \Rightarrow \mathbb{P}_{\mathrm{clean}}^{\mathrm{data}} = \mathbb{P}_{\mathrm{clean}}^{\mathrm{gen}}$. Note that, due to the independence of clean measurements and noise, we have that

$$\hat{\mathbb{P}}_{\mathrm{meas}}^{\mathrm{data}} = \hat{\mathbb{P}}_{\mathrm{clean}}^{\mathrm{data}} \, \hat{\mathbb{P}}_n$$
$$\hat{\mathbb{P}}_{\mathrm{meas}}^{\mathrm{gen}} = \hat{\mathbb{P}}_{\mathrm{clean}}^{\mathrm{gen}} \, \hat{\mathbb{P}}_n. \tag{10}$$

From the assumption that $\hat{\mathbb{P}}_n$ is nonzero everywhere, we deduce that $\hat{\mathbb{P}}_{\mathrm{clean}}^{\mathrm{data}} = \hat{\mathbb{P}}_{\mathrm{clean}}^{\mathrm{gen}}$. This proves the first step.

To prove the next step, we invoke Theorem 4 in [10] which states that any two probability measures $\mathbb{P}_{\mathrm{clean}}^1$ and $\mathbb{P}_{\mathrm{clean}}^2$ that correspond to Dirac probability

measures $\mathbb{P}^1_{\text{conf}}$ and $\mathbb{P}^2_{\text{conf}}$ on $\sum \mathcal{L}_2(\mathbb{R}^3)$, respectively, are mutually singular (zero measure of the common support) if and only if the latter are distinct. We denote the relation of mutual singularity by $\perp$.

Since $\mathbb{P}^{\text{data}}_{\text{conf}}$ is an atomic measure (countable weighted sum of distinct Dirac measures), the corresponding $\mathbb{P}^{\text{data}}_{\text{clean}}$ is composed of a countable sum of mutually singular measures. The same is true for $\mathbb{P}^{\text{gen}}_{\text{clean}}$ since it is equal to $\mathbb{P}^{\text{data}}_{\text{clean}}$.

We proceed by contradiction. We denote by $\text{Supp}\{\mathbb{P}\}$ the support of the measure $\mathbb{P}$. Assume that $\text{Supp}\{\mathbb{P}^{\text{data}}_{\text{conf}}\} \neq \text{Supp}\{\mathbb{P}^{\text{gen}}_{\text{conf}}\}$. Let us define $\mathcal{S}_1 = \text{Supp}\{\mathbb{P}^{\text{gen}}_{\text{conf}}\} \cap \text{Supp}\{\mathbb{P}^{\text{data}}_{\text{conf}}\}^C$. For any $[f] \in \mathcal{S}_1$, we denote by $\mathbb{P}^f_{\text{clean}}$ its noiseless probability measure. Since $f \in \mathcal{S}_1$, it is distinct from any constituent Dirac measure in $\mathbb{P}^{\text{data}}_{\text{conf}}$. Therefore, by using [10, Theorem 4], $\mathbb{P}^f_{\text{clean}}$ is mutually singular to each of the constituent mutually singular measures of $\mathbb{P}^{\text{data}}_{\text{clean}}$, implying that $\mathbb{P}^f_{\text{clean}} \perp \mathbb{P}^{\text{data}}_{\text{clean}}$.

From $\text{Supp}\{\mathbb{P}^f_{\text{clean}}\} \subset \text{Supp}\{\mathbb{P}^{\text{gen}}_{\text{clean}}\}$, it follows that $\mathbb{P}^{\text{data}}_{\text{clean}} \neq \mathbb{P}^{\text{gen}}_{\text{clean}}$, which raises a contradiction. Therefore, the set $\mathcal{S}_1$ is empty. The same can be proved for the set $\mathcal{S}_2 = \text{Supp}\{\mathbb{P}^{\text{gen}}_{\text{conf}}\}^C \cap \text{Supp}\{\mathbb{P}^{\text{data}}_{\text{conf}}\}$. Therefore, $\text{Supp}\{\mathbb{P}^{\text{gen}}_{\text{conf}}\} = \text{Supp}\{\mathbb{P}^{\text{data}}_{\text{conf}}\}$, which means that the location of their constituent Dirac measures are the same. To maintain $\mathbb{P}^{\text{data}}_{\text{clean}} = \mathbb{P}^{\text{gen}}_{\text{clean}}$, the weight of their constituent Dirac measures have to be the same, too. This concludes the proof. $\square$

In essence, Theorem 1 claims that a reconstructed manifold of conformations recovers the true conformations if its measurements match the acquired data in a distributional sense. Though the result assumes the true conformation landscape to be discrete (atomic measure), it holds for an infinite number of discrete conformations which could be arbitrarily close/similar to each other and is thus relevant to continuously varying conformations. We leave the proof of the latter case to future works.

# 7   Conclusion

We have introduced a novel algorithm named Multi-CryoGAN. It can reconstruct the 3D continuous conformation manifold of a protein from a sufficiently rich set of 2D Cryo-EM data. By matching the simulated Cryo-EM projections with the acquired data distribution, Multi-CryoGAN naturally learns to generate a set of 3D conformations in a likelihood-free way. This allows us to reconstruct both continuous and discrete conformations without any prior assumption on the conformation landscape, data preprocessing steps, nor external algorithms such as pose estimation. Our experiments shows that Multi-CryoGAN successfully recovers the molecular conformation manifold, including the underlying distribution. We believe that, with a better incorporation of state-of-the art GAN architectures [13, 12], Multi-CryoGAN could become an efficient and user-friendly method to reconstruct heterogeneous biomolecules in Cryo-EM.

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
