# 8  Supplementary

## Optimization Details

The models are trained end-to-end on the synthetic datasets with the usual WGAN loss (gradient-penalty regularizer $\lambda = 0.6$) on a TITAN X GPU. In all experiments, $G_\gamma, D_\phi$ and the noise parameters are optimized using three separate Adam optimizers with learning rate of $10^{-3}$ and gradient norm clipping value of $1, 10^3, 1$, respectively. Between each generator step, there are $n_{disc} = 5$ discriminator steps. The batch size is kept at 16 samples and the algorithm is run for 30 epochs which was sufficient for the convergence.

## Neural Network Architecture

| LAYER ID | LAYER | RESAMPLE | OUTPUT SHAPE |
|----------|-------|----------|--------------|
| 0 | Input | - | $1 \times 32 \times 32$ |
| 1 | Conv2d | MaxPool | $96 \times 16 \times 16$ |
| 2 | Conv2d | MaxPool | $192 \times 8 \times 8$ |
| 3 | Conv2d | MaxPool | $384 \times 4 \times 4$ |
| 4 | Conv2d | MaxPool | $768 \times 2 \times 2$ |
| 5 | Flatten | - | $3072 \times 1 \times 1$ |
| 6 | FC | - | $50 \times 1 \times 1$ |
| 7 | FC | - | $1 \times 1 \times 1$ |

Table 1: 2D Discriminator architecture. LeakReLU(0.1) is used after every Max-Pool and in layer 6.

| Layer id | Layer | Resample | Norm | Output Shape (C, D, H, W) |
|---|---|---|---|---|
| 0 | Input | - | - | $1 \times 32 \times 32 \times 32$ |
| 1 | Conv3d | - | BN | $16 \times 32 \times 32 \times 32$ |
| 2 | Conv3d | MaxPool | BN | $16 \times 16 \times 16 \times 16$ |
| 3 | Conv3d | - | BN | $32 \times 16 \times 16 \times 16$ |
| 4 | Conv3d | MaxPool | BN | $32 \times 8 \times 8 \times 8$ |
| 5 | Conv3d | - | BN | $64 \times 8 \times 8 \times 8$ |
| 6 | Conv3d | MaxPool | BN | $64 \times 4 \times 4 \times 4$ |
| 7 | Conv3d | - | BN | $128 \times 4 \times 4 \times 4$ |
| 8 | Conv3d | MaxPool | BN | $128 \times 2 \times 2 \times 2$ |
| 9 | Conv3d | - | BN | $256 \times 2 \times 2 \times 2$ |
| 10 | Conv3d | - | BN | $256 \times 2 \times 2 \times 2$ |
| 11 | Conv3d | Upsample | BN | $128 \times 4 \times 4 \times 4$ |
| 12 | Concat(layer 8) | - | - | $256 \times 4 \times 4 \times 4$ |
| 13 | Conv3d | - | BN | $128 \times 4 \times 4 \times 4$ |
| 14 | Conv3d | - | BN | $128 \times 4 \times 4 \times 4$ |
| 15 | Conv3d | Upsample | BN | $64 \times 8 \times 8 \times 8$ |
| 16 | Concat(layer 6) | - | - | $128 \times 8 \times 8 \times 8$ |
| 17 | Conv3d | - | BN | $64 \times 8 \times 8 \times 8$ |
| 18 | Conv3d | - | BN | $64 \times 8 \times 8 \times 8$ |
| 19 | Conv3d | Upsample | BN | $32 \times 16 \times 16 \times 16$ |
| 20 | Concat(layer 4) | - | - | $64 \times 16 \times 16 \times 16$ |
| 21 | Conv3d | - | BN | $32 \times 16 \times 16 \times 16$ |
| 22 | Conv3d | - | BN | $32 \times 16 \times 16 \times 16$ |
| 23 | Conv3d | Upsample | BN | $16 \times 32 \times 32 \times 32$ |
| 24 | Concat(layer 2) | - | - | $32 \times 32 \times 32 \times 32$ |
| 25 | Conv3d | - | BN | $16 \times 32 \times 32 \times 32$ |
| 26 | Conv3d | - | BN | $16 \times 32 \times 32 \times 32$ |
| 27 | Conv3d | - | BN | $1 \times 32 \times 32 \times 32$ |

Table 2: 3D Generator architecture. ReLU is used after every BatchNorm (BN). Concatenation is with the values before pooling.