# OpenReview forum: "Multi-CryoGAN: Reconstruction of Continuous Conformations in Cryo-EM  Using Generative Adversarial Networks"
_thecvf.com/ECCV/2020/Workshop/BIC — BIC 2020 Oral_

### Official Review · AnonReviewer2 · 2020-07-30
**Original and well described method for multi-conformation Croy-EM reconstruction**

**Rating:** 9
**Confidence:** 4

**Review:**

## Summary

The paper builds on an existing method (CryoGAN) that reconstructs a 3D density of a biomolecule from random 2D tomographic Cryo-EM projections and extends it to the case when the biomolecule is present in multiple structural conformations. Specifically, the authors propose to use a generator module G(z) over a latent space z that parameterized a distribution of possible 3D molecule densities. The sampled densities  together with a forward model produce synthetic 2D tomographic projections which are together with real 2D projection are used to train a discriminator/critic as part of a Wasserstein GAN like approach. The paper then demonstrates that for a synthetic dataset that both with a discrete and a continuous conformation space can be recovered.

## Strengths

- Approach is original and interesting
- Provides strong theoretical foundation of the method

## Weaknesses

- Description of latent code -> conformation manifold could be made clearer

## Overall

The paper is clearly written, the presented approach is interesting and well described, and the results are convincing. The approach of using a GAN like generator in conjunction with a realistic forward model that generates 2D tomographic samples from the 3D density and then to train a critic on the latter (i.e. CryoGAN) seems to be a very powerful and flexible method.  Just a few issues:

- As far as I can tell (cf Algorithm 2), the actual training scheme is simply a WassersteinGAN (with a differentiable forward model on top of the generator), or? That could be made a little bit clearer in the text.
- For the latent code  "pz ~ Uniform (z0 , z1 ), where z0, z1 in R^{32,32,32} are randomly chosen from Uniform(0,0.025)"
Why was that specific initialization chosen?
- Meaning of the latent space: In the paper the latent code is affinely mapped to the conformation space via (1-a)*z1 + a*z2 . Why would that be the correct mapping? As far as I understood the latent space is of size 32x32x32 (is that correct?) and there must be many latent paths that would correspond to a path in conformation space. If the conformation space is now 2 or 3 dimensional, which subset of the latent space would one choose then? What would happen, if the conformation manifold does not even admit a single global chart (e.g. if the molecule has a subunit that can freely rotate around an angle 0...2pi)?


## Minor comments

- typos: "carries", "space f", "versiond"
- "called a hypermolecule, which is characterized by a basis of hypercomponents" -> that's a bit hard to understand.
- "they rely on 3D clustering to deal with structural variations of protein complexes" -> why 3D clustering? In which space (poses, conformations) is clustered?
- line 402: is there alpha missing before z_1?
- what would be if orientations are not uniformly distributed on SO3? Would the reconstruction simply be worse?







**Reviews Visibility:**

I agree that my anonymized review is made publicly visible, if the submission is accepted.

---

### Official Review · AnonReviewer1 · 2020-07-31
**Interesting and potentially very useful extension of existing cryoGAN work. Solid paper!**

**Rating:** 8
**Confidence:** 4

**Review:**

The authors present multi-cryoGAN, a generative adversarial neural network for reconstruction of continuous molecular conformations. The method is an extension of cryoGAN, which can perform the reconstruction task for single conformations only. Understanding the continuous motion between individual conformations is important for the understanding of biological mechanisms. The authors evaluate their method on a synthetic dataset, once for a continuous range of conformations and once for two discrete conformational states.

# Strenghts
The practical value of such a method is very high and there is certainly demand in the field.
The related work section nice reviews existing work in this field.

# Weaknesses
The paper is missing a comparison to other state-of-the-art methods like cryoDRGN (https://www.biorxiv.org/content/10.1101/2020.03.27.003871v1), which is mentioned in the introduction but not compared to.
It would be interesting to see the FSC curves for reconstruction from different methods, even if they are only able to reconstruct a single conformation.

# Comments
* In the theory part the authors mention the possibility of reconstructing conformations with translation/shift errors i.e. not perfectly picked locations. Later, in the experiments, this option is disabled, therefore not testing the ability of the proposed method to handle translations/shifts.
* In the experiments only Gaussian noise is used. It would nice if also Poisson noise would have been used. Is there any problems to expect when switching to real data (and the associated noise sources)?
* Figure 4a: it would be quite interesting to see the resolution of the reconstructed particles for each different conformation at 0.5 FSC.
* Figure 5a: the reconstruction seems to have a tendency to prefer the two extreme conformations -- why do you think that is? And how would an experiment of discrete conformations with a 50-50 split look like in comparison?
* Will the code for this work be available online? If so, the authors should point to it in a footnote or so...

**Reviews Visibility:**

I agree that my anonymized review is made publicly visible, if the submission is accepted.

---

### Decision · Program_Chairs · 2020-07-31

Accept (Oral)